# Galectin-3-ITGB1 Signaling Mediates Interleukin 10 Production of Hepatic Conventional Natural Killer Cells in Hepatitis B Virus Transgenic Mice and Correlates with Hepatocellular Carcinoma Progression in Patients

**DOI:** 10.3390/v16050737

**Published:** 2024-05-07

**Authors:** Yongyan Chen, Wendi Zhang, Min Cheng, Xiaolei Hao, Haiming Wei, Rui Sun, Zhigang Tian

**Affiliations:** 1Key Laboratory of Immune Response and Immunotherapy, Institute of Immunology, Biomedical Sciences and Health Laboratory of Anhui Province, Center for Advanced Interdisciplinary Science and Biomedicine of IHM, School of Basic Medical Sciences, Division of Life Sciences and Medicine, University of Science and Technology of China, Hefei 230027, China; wendiz@mail.ustc.edu.cn (W.Z.); chengmin@ustc.edu.cn (M.C.); ustc.bio.hxl@gmail.com (X.H.); ustcwhm@ustc.edu.cn (H.W.); sunr@ustc.edu.cn (R.S.); 2Research Unit of NK Cell Study, Chinese Academy of Medical Sciences, Hefei 230027, China; 3Department of Geriatrics, Gerontology Institute of Anhui Province, The First Affiliated Hospital of USTC, Division of Life Sciences and Medicine, University of Science and Technology of China, Hefei 230001, China

**Keywords:** HBV, natural killer cell, galectin-3, ITGB1, interleukin 10

## Abstract

Background and Aims: The outcomes of HBV infections are related to complex immune imbalances; however, the precise mechanisms by which HBV induces immune dysfunction are not well understood. Methods: HBV transgenic (HBs-Tg) mice were used to investigate intrahepatic NK cells in two distinct subsets: conventional NK (cNK) and liver-resident NK (LrNK) cells during a chronic HBV infection. Results: The cNK cells, but not the LrNK cells, were primarily responsible for the increase in the number of bulk NK cells in the livers of ageing HBs-Tg mice. The hepatic cNK cells showed a stronger ability to produce IL-10, coupled with a higher expression of CD69, TIGIT and PD-L1, and lower NKG2D expression in ageing HBs-Tg mice. A lower mitochondrial mass and membrane potential, and less polarized localization were observed in the hepatic cNK cells compared with the splenic cNK cells in the HBs-Tg mice. The enhanced galectin-3 (Gal-3) secreted from HBsAg^+^ hepatocytes accounted for the IL-10 production of hepatic cNK cells via ITGB1 signaling. For humans, *LGALS3* and *ITGB1* expression is positively correlated with *IL-10* expression, and negatively correlated with the poor clinical progression of HCC. Conclusions: Gal-3-ITGB1 signaling shapes hepatic cNK cells but not LrNK cells during a chronic HBV infection, which may correlate with HCC progression.

## 1. Introduction

Chronic hepatitis B virus (HBV) infection and HBV-related HCC are global public health problems, with a high prevalence and incidence in Asia and, particularly, in China [1,2]. As an attractive target site for HBV, the liver is an immunological organ with numerous populations of immune cells and predominantly with the innate immune system; the balance between immunity and tolerance and dynamic interactions between these immune cells are essential to liver function and tissue health [3]. The outcomes of HBV infections are related to the magnitude and quantity of the host’s anti-HBV immune responses and a complex immune imbalance [4,5]. The activation and exhaustion of CD4^+^ T, CD8^+^ T, natural killer (NK), natural killer T (NKT), monocytes/macrophages and hepatic stellate cells (HSCs) that participate in liver inflammation and eventually HCC development due to a chronic HBV infection, and the immunosuppressive cells, such as the myeloid-derived suppressor cells (MDSCs), regulatory T cells (Tregs), regulatory B cells (Bregs) and Kupffer cells (KCs), that regulate anti-HBV and antitumor responses, result in complicated crosstalk and immune disorders [5]. However, the precise mechanisms by which HBV induces immune dysfunction are not well understood and require further investigation.

In the liver, enriched NK cells exhibit early antiviral and antitumor activities, with a high percentage of 30–50% of the intrahepatic lymphocytes in humans and 5–10% of the intrahepatic lymphocytes in mice [6,7]. During a chronic HBV infection, HBV-infected hepatocytes express high levels of NKG2D ligands, and NK cell hepatocytotoxicity is induced in a NKG2D-ligand interaction-dependent manner [8,9]. NK cells display anti-HBV roles and mediate liver inflammation, which promotes the development of HCC by producing IFN-γ [10,11]; meanwhile, their antiviral activities are suppressed by regulatory cells and cytokines, and feature upregulated inhibitory receptors NKG2A, TIM-3 and PD-1, and reduced IFN-γ and TNF-α production [11,12,13]. Previous studies have elucidated the roles of bulk NK cells in the liver; however, hepatic NK cells harbor two distinct subsets—CD49a^−^CD49b^+^ conventional NK (cNK) cells and CD49a^+^CD49b^−^ liver-resident NK (LrNK) cells—which have different origins, phenotypes and functions in a steady state [14,15].

LrNK cells are also called type 1 innate lymphoid cells (ILC1s); thus, cNK cells and LrNK/ILC1s are referred to as group 1 ILCs [16]. Hepatic cNK cells that are generated from NK cell precursors (NKPs) are dependent on the transcription factors NFIL3 and Eomes, and emerge 2–3 weeks after birth. LrNK cells emerge before birth and develop from ILC precursors (ILCPs), with strict requirements for the transcription factors T-bet and Hobbit, but not Eomes [17,18,19,20]. An IFN-γ-dependent loop that promotes the development of LrNK cells in situ, but not cNK cells, has been identified [14]. LrNK cells can produce larger amounts of TNF-α, but smaller amounts of IFN-γ and perforin, and similar amounts of granzyme B when compared with cNK cells [14]. In a mouse model mimicking an acute HBV infection, hepatic cNK cells, but not LrNK/ILC1s, were involved in promoting the antiviral activity of CD8^+^ T cells by secreting IFN-γ [21]. Elevated ILC1s were significantly related to hepatic damage in CHB patients, indicating the potential proinflammatory roles of ILC1s in the pathogenesis of CHB [22]. Thus, the potential roles of cNK and LrNK cells in the maintenance of liver homeostasis and the progression of liver diseases and cancer deserve further investigation, and will contribute to the proposal of effective treatments for CHB and HBV-related HCC.

The liver tissue microenvironment shapes intrahepatic lymphocytes and influences their functions [23]. Galectin-3 (Gal-3), a β-galactoside-binding lectin, is located extracellularly, on the cell surface and intracellularly, and can bind to N-glycans on extracellular matrix proteins, cell receptors and pathogens. In the liver, Gal-3 is involved in inducing liver injury, liver steatosis, bile duct damage, liver fibrosis and HCC [24,25]. Gal-3 strongly antagonizes human NK cell attacks against tumors in vivo, indicating its inhibition of NK cells [26]. Gal-3 is expressed by proliferating or transformed hepatocytes, but is absent in normal hepatocytes [27]. The prognostic significance of serum Gal-3 levels has been determined in patients with HCC and chronic HBV infections [28,29]; however, its precise roles in the liver microenvironment are not well elucidated.

In this study, HBs-Tg mice, which are a model for chronic HBV carriers who develop malignant transformation and spontaneous HCC, were used to explore intrahepatic NK cells, including cNK and LrNK cells [10,30]. It is demonstrated that enhanced Gal-3 secreted from HBsAg^+^ hepatocytes induces IL-10 production in hepatic cNK cells, but not in LrNK cells, via ITGB1 signaling, which correlates with HCC progression. These findings indicate the precise mechanisms by which HBV induces the immune dysfunction of NK cells in the liver microenvironment, which will be helpful for controlling HBV-related liver disease.

## 2. Materials and Methods

### 2.1. Animals

Eight- to ten-week-old male HBV transgenic mice C57BL/6J-TgN (AlblHBV) 44Bri (named HBs-Tg mice) were purchased from the Department of Laboratory Animal Science of Peking University (Beijing, China). C57BL/6J littermates were used as the controls. Four- to five-month-old and twelve- to thirteen-month-old male HBs-Tg mice and their littermates were obtained for this study. All the mice were kept under specific, pathogen-free controlled conditions (22 °C, 55% humidity and 12 h day/night rhythm) in compliance with the guidelines outlined in the Guide for the Care and Use of Laboratory Animals.

### 2.2. Reagents

Mitochondrion-selective probes, including MitoTracker^®^ Green FM (Invitrogen, Carlsbad, CA, USA) and MitoTracker^®^ Orange CMTMRos (Invitrogen, Carlsbad, CA, USA), were used to label the mitochondria of the NK cells. TD139 (TargetMol, Wellesley Hills, MA, USA) was dissolved in dimethyl sulfoxide (DMSO) (Solarbio, Beijing, China) and used for the inhibition of Galectin-3 in vivo (15 μg/g body weight, once per 24 h for three times). A Galectin-3 mouse ELISA Kit (Invitrogen, Carlsbad, CA, USA) was used to determine the serum and liver homogenate levels of Galectin-3. A mouse IL-10 ELISA kit (Abcam, Cambridge, MA, USA) was used to determine the serum and liver homogenate levels of IL-10.

### 2.3. Mononuclear Cell (MNC) Preparation

Briefly, the liver cells were resuspended in 40% Percoll solution (GIBCOL BRL, Grand Island, NY, USA) and the cell mixture was gently overlaid onto 70% Percoll solution and then centrifuged at 750× *g* for 30 min at room temperature. Splenocytes were treated with RBC lysis solution (BD PharMingen, San Diego, CA, USA).

### 2.4. Purification of cNK Cells

The liver cNK cells were purified by a mouse CD49b Positive Selection Kit (Stem Cell, Vancouver, BC, Canada), and the splenic cNK cells were purified by a mouse NK Cell Isolation Kit (Miltenyi Biotec, Auburn, CA, USA). The LrNK cells (CD3^−^NK1.1^+^CD49b^−^CD49a^+^) and the cNK cells (CD3^−^NK1.1^+^CD49b^+^CD49a^−^) were purified by fluorescence-activated cell sorting (FACS) (BD Aria II, BD Biosciences, San Jose, CA, USA).

### 2.5. Flow Cytometry Analysis

After blocking with normal rat serum to saturate the Fc receptors, the MNCs were stained with a saturating amount of the fluorescence-labeled mAbs for the surface antigens at 4 °C for 30 min in darkness. For the intracellular cytokine assay, freshly isolated MNCs were stimulated with 30 ng/mL PMA (Sigma, St Louis, MO, USA) and 1 μg/mL ionomycin (Sigma, St Louis, MO, USA), and treated with 10 μg/mL monensin (Sigma, St Louis, MO, USA) for 4 h. Further, after staining the surface antigens, the cells were fixed and permeabilized using 100 μL of Cytofix and Cytoperm solution (eBioscience, San Diego, CA, USA). A Fortessa cytometer (BD Biosciences, San Jose, CA, USA) was used and the data were analyzed using FlowJo (Tree Star, BD Biosciences, San Jose, CA, USA). The antibodies used are shown in the Appendix A and methods.

### 2.6. Electron Microscopic Morphology of NK Cells

Purified LrNK cells (CD3^−^NK1.1^+^CD49b^−^CD49a^+^) and cNK cells (CD3^−^NK1.1^+^CD49b^+^CD49a^−^) were obtained by FACS. The cell samples were fixed with 2.5% glutaraldehyde and 1% osmic acid and embedded with Epon812 epoxy resin, and then sliced into ultraslices of 50–70 nm and stained with 2% uranyl acetate and 6% lead citrate to observe their ultrastructure under a Tecnai G^2^ 120 kV transmission electron microscope (FEI, Boston, MA, USA).

### 2.7. Serum Transaminase Activity Assays

Serum alanine aminotransferase (ALT) activities were determined by the standard photometric method, using a serum transaminase test kit (Rong Sheng, Shanghai, China).

### 2.8. Galectin-3 Binding Analysis

The MNCs were collected, to which was added 1 μg/mL recombinant mouse galectin-3 protein (R&D Systems, Minneapolis, MN, USA), and then cultured in PBS at 4 °C for 30 min. The MNCs were harvested and washed with PBS, and then examined for Gal-3 binding by a Fortessa cytometer (BD Biosciences, San Jose, CA, USA).

### 2.9. Galectin-3 Stimulation In Vitro

The MNCs (5 × 10^5^) were plated into 48-well plates (400 μL/well), and the purified cNK cells (1 × 10^5^) were plated into 96-well plates (200 μL/well) and cultured with 10% FBS 1640 medium supplemented with 100 U/mL IL-2. Recombinant mouse galectin-3 protein (R&D Systems, Minneapolis, MN, USA) was added to stimulate the cells for 48 h in vitro (2.5 μg/mL, 5 μg/mL or 50 ng/mL). Monensin (Sigma, St Louis, MO, USA) was used at a concentration of 2.5 μg/mL to block the secretion of IL-10 by the cultured NK cells in vitro. The culture supernatant was harvested for IL-10 detection by a mouse IL-10 Cytometric Bead Array (CBA) Flex set (BD Biosciences, San Diego, CA, USA) and Mouse Solbl Ptein CBA Buf Kit (BD Biosciences, San Diego, CA, USA). Anti-CD29 (30 μg/mL, clone Ha2/5, BD Pharmingen, CA, USA) was used to block the interaction of Galectin-3 with ITGB1 (CD29) on the NK cells. Control IgG (30 μg/mL, clone Ha4/8, BD Pharmingen, CA, USA) was used.

### 2.10. Correlation Analysis

Gene Expression Profiling Interactive Analysis (GEPIA2) (http://gepia2.cancer-pku.cn/#correlation (GEPIA 2—Copyright © 2018 Zhang’s Lab)) was used to perform the correlation analysis for the given sets of TCGA and GTEx expression data. The correlation coefficient was analyzed using Spearman. Liver hepatocellular carcinoma (LIHC) normal (50 samples) and tumor (369 samples) samples, and GTEx liver (110 samples) samples were included.

### 2.11. Statistical Analysis

The results were analyzed using Student’s *t* test or analysis of variance (ANOVA) as appropriate. All the data are shown as the mean ± SEM. *p* < 0.05 was considered significant.

## 3. Results

### 3.1. Increased Hepatic cNK Cells Were Skewed to Produce IL-10 and Exhibited Immune Inhibitory Features with Ageing in HBs-Tg Mice

The mononuclear cells (MNCs) in the liver and spleen were analyzed by flow cytometry to show the NK cell immune responses in HBs-Tg mice compared with wild-type (WT) B6 mice (Appendix A). No significant differences were observed in the number of NK cells in the livers of the 8- to 10-week-old HBs-Tg mice and the control WT B6 mice and the 4- to 5-month-old HBs-Tg mice (Figure 1A). However, a significantly increased number of NK cells in the liver was observed in the 12- to 13-month-old HBs-Tg mice compared with the WT B6 mice (Figure 1A). Among these cells, the CD49b^+^ cNK cells, but not the CD49a^+^ LrNK cells, were primarily responsible for the increase in the number of bulk NK cells in the livers of the HBs-Tg mice (Figure 1A). These alternations existed in the liver but not in the spleen, since no differences were observed in the number of NK cells in the spleen during the lifespan of the HBs-Tg mice and WT B6 mice (Figure 1B).

Interestingly, the intrahepatic NK cells showed a stronger ability to produce IL-10 in the 12- to 13-month-old HBs-Tg mice when compared with the control B6 mice (Figure 2A), but not the splenic NK cells (Figure 2B). No differences in IFN-γ production were observed (Figure 2A,B). Furthermore, the intrahepatic cNK cells showed higher expression levels of IL-10 than the splenic cNK cells in the HBs-Tg mice, and the LrNK cells did not exhibit the ability to produce IL-10 in the HBs-Tg mice, which were the same as in the control B6 mice (Figure 2C). These results indicate that the intrahepatic cNK cells, but not the LrNK cells, account for IL-10 production in aged HBs-Tg mice. Furthermore, these intrahepatic cNK cells showed higher expression levels of CD69, TIGIT and PD-L1, and lower expression levels of NKG2D in the aged HBs-Tg mice compared with the control B6 mice (Figure 2D), but no significant differences were observed in the splenic cNK cells between the HBs-Tg mice and control B6 mice (Figure 2D). The phenotype changes in the cNK cells were not observed in the liver and spleen of the adult HBs-Tg mice (8- to 10-week old) or the control B6 mice (Appendix A). These results indicate that hepatic cNK cells are skewed to produce IL-10 and exhibit immune inhibitory features with the ageing of HBs-Tg mice.

### 3.2. Hepatic cNK Cells Showed a Distinct Mitochondrial Signature in HBs-Tg Mice

In conventional lymphocytes, mitochondria play a central role in the coordination of cell fate decisions and functional immune responses. We used MitoTracker Green FM and Orange CMTMRos staining to assess the mitochondrial mass and membrane potential of the hepatic cNK cells compared with the splenic cNK cells and LrNK cells. The mitochondrial mass of the hepatic cNK cells was significantly lower than that of the splenic cNK cells in the HBs-Tg mice (Figure 3A), and the mitochondrial membrane potential of the hepatic cNK cells was also significantly lower (Figure 3B). There were no significant differences in mitochondrial mass or membrane potential between the hepatic cNK and LrNK cells (Figure 3A,B). Further, we studied the mitochondrial morphology and confirmed that the hepatic cNK cells had a lower mitochondrial mass and a less polarized localization than that observed in the splenic cNK cells of the HBs-Tg mice, as shown by the dispersed mitochondria of a smaller size (Figure 3C). These results indicate that exposure to liver environmental signals might be the key element affecting cNK cells during HBV infection, leading to the differences between the hepatic cNK cells and splenic cNK cells.

### 3.3. HBsAg^+^ Hepatocytes Enhanced Galectin-3 Expression in HBs-Tg Mice

The hepatocytes from the 4- to 5-month-old HBs-Tg mice and control B6 mice were purified and then analyzed through mRNA sequencing. There were 451 DEGs between the HBs-Tg and B6 mice, with 295 upregulated genes (65.41%) and 156 downregulated genes (34.59%) (Figure 4A). A GO enrichment analysis showed that these DEGs in the HBsAg^+^ hepatocytes were mainly negatively enriched in metabolic process and biosynthetic process, and positively enriched in chemotaxis, the regulation of inflammatory responses and cellular responses to cytokines (Figure 4B). The top-50 DEGs showed that HBsAg^+^ hepatocytes might be involved in immune responses (Figure 4C). Given the cell interaction, the HBsAg^+^ hepatocytes of the HBs-Tg mice expressed higher levels of *Lgals3* compared with the control B6 mice (Figure 4D), which was further confirmed by a real-time PCR (Figure 5A). There were no significant differences in the serum levels of Gal-3 between the HBs-Tg mice and the control B6 mice (Figure 5B), but much higher protein levels of Gal-3 were observed in the liver tissues of the HBs-Tg mice compared with the control B6 mice (Figure 5C). Furthermore, the protein levels of Gal-3 in the liver tissues increased in the 4- to 5-month-old HBs-Tg mice compared with the 8- to 10-week-old HBs-Tg mice (Figure 5C). By a histochemistry analysis, the enhanced expression of Gal-3 was observed in the HBsAg^+^ hepatocytes of the HBs-Tg mice (Figure 5D).

### 3.4. Galectin-3 Accounted for IL-10 Production in Hepatic cNK Cells and Prevented Liver Injury in HBs-Tg Mice

To assess the effects of enhanced Gal-3 on NK cells, we determined the expression of IL-10 in NK cells treated with recombinant mouse galectin-3 (rmGal-3) by in vitro experiments. Gal-3 significantly increased IL-10 expression in the cNK cells, especially the hepatic cNK cells from the HBs-Tg mice (Figure 6A). Enhanced levels of IL-10 were detected in the culture supernatant after Gal-3 stimulation for 48 h (Figure 6B). The secretion of IL-10 from the cNK cells could be blocked by a monensin treatment, as shown by the significantly reduced level of IL-10 in the culture supernatant of the monensin-treated group, further indicating that Gal-3 mediated IL-10 production in the hepatic cNK cells (Figure 6C).

Further, we assessed the role of Gal-3 by in vivo experiments using a Gal-3 inhibitor (TD139) treatment. The inhibition of Gal-3 significantly decreased the serum levels of IL-10 in the HBs-Tg mice (Figure 6D). In the livers of the HBs-Tg mice, IL-10-positive cells could be detected, but in the livers of the TD139-treated HBs-Tg mice, no IL-10-positive cells were observed (Figure 6E), which further confirms the induction of IL-10 secretion by Gal-3. In the livers of the TD139-treated HBs-Tg mice, cNK cells were found to express higher levels of the activating receptor NKG2D, indicating that high levels of Gal-3 in the liver inhibited their activation (Figure 6F). Accordingly, hepatocyte injury was induced when Gal-3 was inhibited, as demonstrated by the increased serum levels of ALT in the TD139-treated HBs-Tg mice (Figure 6G).

These results indicate that Gal-3 was responsible for IL-10 production in the hepatic cNK cells and prevented liver injury in the HBs-Tg mice. To gain insight into the clinical relevance of this for humans, a correlation analysis was performed by using the TCGA and GTEx expression data. Significant correlations between the *LGALS3* and *IL-10* mRNA levels were observed in the LIHC normal and tumor tissues, and also in the normal liver tissues (Appendix A).

### 3.5. Galectin-3 Induced IL-10 Transcription via ITGB1 Signaling in Hepatic cNK Cells

Extracellular Gal-3 can interact with several kinds of receptors with glycoconjugates on their cell surfaces to mediate cell signaling, such as CD107a, lymphocyte-activation gene-3 (LAG-3) and integrin subunit beta 1 (ITGB1). Here, we found low expression levels of CD107a and LAG-3 (Appendix A), but high expression levels of ITGB1 (CD29) in the hepatic cNK cells in the HBs-Tg mice (Appendix A), indicating the possibility of an interaction between Gal-3 and CD29. As shown in Figure 7A,B, in the HBs-Tg mice, the hepatic cNK cells express much higher levels of CD29 compared with the splenic cNK cells and LrNK cells. Correspondingly, a higher Gal-3 binding rate with hepatic cNK cells than splenic cNK cells is confirmed (Figure 7C). Further, whether Gal-3 induces IL-10 production in hepatic cNK cells through CD29 was determined. Gal-3 strongly induced IL-10 transcription (>20-fold) in the hepatic cNK cells after 48 h of stimulation, as detected by a real-time PCR (Figure 7D). Treatment with a neutralizing antibody against ITGB1 (anti-CD29) dramatically inhibited IL-10 transcription (Figure 7D). Gal-3 did not affect the cytotoxicity of the cNK cells against tumor target cells (Appendix A). These results demonstrate that Gal-3 induced IL-10 production in the hepatic cNK cells via ITGB1 signaling in the HBs-Tg mice.

### 3.6. LGALS3 and ITGB1 Expression Negatively Correlated with the Poor Progression of HCC

Having identified the important role of Gal-3 derived from HBsAg^+^ hepatocytes by using HBs-Tg mice, we speculated that enhanced Gal-3 levels might correlate with HBV-related HCC development. According to the TGCA human database, higher expression levels of *LGALS3* are observed in HCC tumor tissues than in non-tumor tissues, and the expression levels are positively correlated with the stages of HCC (Figure 8A,B). *LGALS3* expression is negatively correlated with the overall survival of HCC patients (Figure 8C). Significant correlations between *ITGB1* and *LGALS3* mRNA levels, and between *ITGB1* and *IL-10* mRNA levels, are observed in the LIHC normal and tumor tissues, and in normal liver tissues (Appendix A). As a result, *ITGB1* expression is negatively correlated with the overall survival of HCC patients (Figure 8D). These results indicate the tumor-promoting role of Gal-3-ITGB1 signaling in HCC development.

## 4. Discussion

The involvement of NK cell dysfunction in chronic HBV infection and HBV-related HCC prompted us to further explore the exact roles of cNK and LrNK cells in the liver, since these two distinct NK cell subsets have been identified with different functions. In this study, we found that hepatic cNK cells were skewed to produce IL-10, but not the LrNK cells in ageing HBs-Tg mice, and exhibited suppressive features and distinct mitochondrial signatures. Enhanced Gal-3 secreted from HBsAg^+^ hepatocytes accounted for the IL-10 transcription in the hepatic cNK cells via ITGB1 signaling. Our findings elucidate a novel mechanism for the interactions between HBV^+^ hepatocytes and cNK cells, by which HBV initiates immunosuppressive cNK cells in the liver, contributing to HBV persistence and disease progression.

Previous studies have shown that HBV-promoting IL-10 in the microenvironment is derived from macrophages, Tregs or dendritic cells (DCs) [31,32]. Notably, peripheral-circulating NK cells have been demonstrated to express higher levels of IL-10 in chronic HBV patients than in healthy individuals, and to function as regulators with suppressive features, and are educated by HBsAg-induced suppressive monocytes via PD-L1/PD-1 and HLA-E/CD94 signals [33]. Here, we demonstrate that intrahepatic cNK cells, but not LrNK cells, are shaped to produce IL-10 by HBsAg^+^ hepatocytes via Gal-3-ITGB1 signaling (Figure 5, Figure 6 and Figure 7). In humans, the different relationships between serum HBV pgRNA and CD56^dim^ NK cells and CD56^high^ NK cells have been explored, indicating the differences in these NK cell subsets [34]. However, human LrNK cells remain challenging to identify, for example, CD49a^+^ CD56^bright^ NK cells, CXCR6^+^ NK cells and CD49e^−^ NK cells, so that the roles of LrNK cells in chronic HBV infection and related liver disease are elusive. Recently, CD49a^+^ NK cells were reported to be enriched in HCC, and a more abundant infiltration was present in patients in advanced stages [35]. Additionally, the presence of mature, HBV-specific memory NK cells (mNKs) was demonstrated in humans exposed to HBV antigens through vaccination or infection, which was mediated by CD56^dim^ NK cells co-expressing CD57, CD69 and KLRG1 [36]. Thus, the alternations in distinct NK cell subsets during HBV infection are interesting issues and their relation to HBV-related liver diseases deserve further investigation.

As an important immune regulator, Gal-3 modulates NK cells, since the complete absence of Gal-3 in an endogenous host results in decreased NK cytotoxicity and a disturbed Th1/Th2/Th17 cytokine milieu balance [37]. Conversely, a lack of Gal-3 enhances the percentage of effective cytotoxic CD27^high^ CD11b^high^ NK cells, as well as immature CD27^high^ CD11b^low^ NK cells, and promotes NK-mediated antitumor responses [38]. Tumor-released Gal-3 contributes to tumor escape from NK cell attack as a soluble inhibitory ligand, by interacting with receptor NKp30 in humans [26]. However, Gal-3 did not affect the cytotoxicity of hepatic cNK cells against tumor target cells in either the B6 mice or the HBs-Tg mice (Figure 5), which might be explained by the lack of NKp30 receptors on mouse NK cells, since NKp30 is only a pseudogene in mice [39]. Additionally, Gal-3 reduces the affinity of MICA for its receptor NKG2D, and severely impairs NK cell activation, thereby promoting tumor evasion [40]. In this study, we demonstrate a new mode of Gal-3 interaction with NK cells, in which HBsAg^+^ hepatocyte-derived Gal-3 can interact with its receptor ITGB1 on the cNK cells and induce their IL-10 production (Figure 7). Gal-3 activates pancreatic stellate cells to produce the inflammatory cytokine IL-8 via ITGB1 signaling, thereby promoting the growth and metastasis of pancreatic orthotopic tumors in mice [41]. Thus, the roles of Gal-3 are closely related to certain environments.

Gal-3 has been closely associated with a poor prognosis of primary HCC, by promoting tumor cell growth, migration and invasion, stimulating angiogenesis and enhancing the tumorigenesis and metastasis of HCC cells [24,42,43]. After persistent HBV infection, Gal-3 was demonstrated to play a key role in adverse disease progression and to be a potential therapeutic target [44]. In this study, LGALS3 expression was negatively correlated with the overall survival of HCC patients (Figure 8C), which was consistent with the levels of *ITGB1* and *IL-10* (Appendix A). These results indicate that Gal-3 may play a role in HCC development by modulating NK cell function, in addition to regulating HCC tumor cells. Proliferating cells in the process of being transformed express Gal-3, indicating an early neoplastic event [27]. In the HBs-Tg mice, HBsAg^+^ hepatocytes expressed higher levels of Gal-3 compared with the WT B6 mice (Figure 5), consistent with the findings of epithelial-to-mesenchymal transition (EMT) in HBsAg^+^ hepatocytes [10]. However, Gal-3^−/−^ mice have developed dysplastic liver nodules and HCC [45]. The antitumor activity of the carbohydrate recognition domain of Gal-3 was demonstrated in HCC by inhibiting cell viability, migration and invasion of tumor cells [46].

In this study, a new mode of mechanism was elucidated, in which enhanced Gal-3 secreted from HBsAg^+^ hepatocytes induced IL-10 production in hepatic cNK cells, but not in LrNK cells, via ITGB1 signaling, which correlated with HCC progression. These findings indicate the precise roles of intrahepatic NK cells, including cNK and LrNK cells, during chronic HBV infection, which will be helpful for controlling HBV-related liver disease.

## Figures and Tables

**Figure 1 viruses-16-00737-f001:**
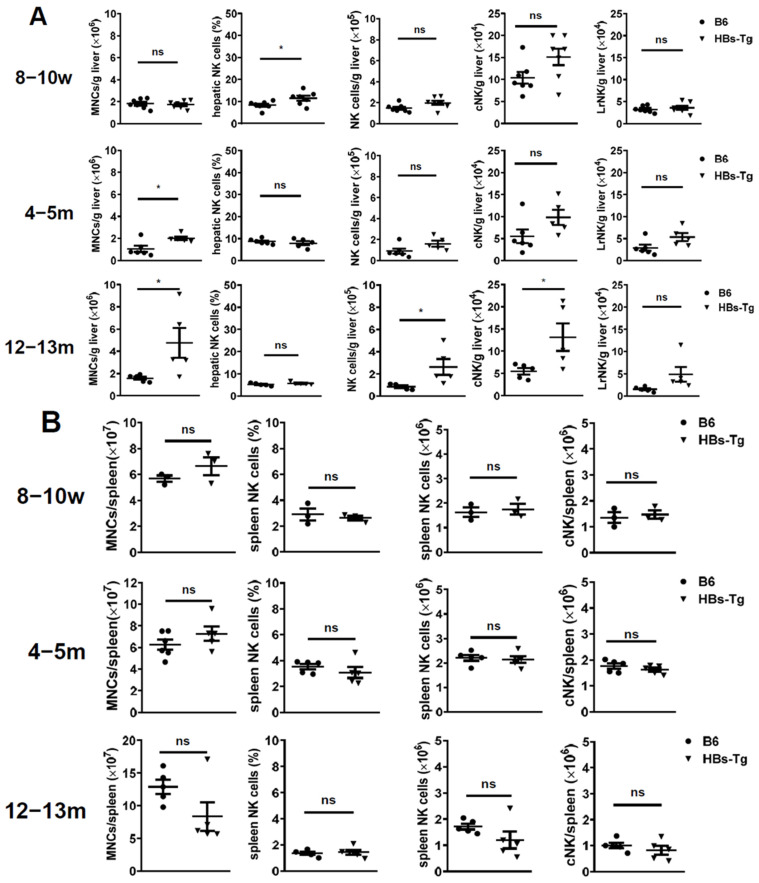
The number of cNK cells, but not LrNK cells, was significantly increased in the livers of 12- to 13-month-old HBs-Tg mice. MNCs were isolated from the livers and spleens of 8- to 10-week-old, 4- to 5-month-old and 12- to 13-month-old HBs-Tg mice and the control WT B6 mice, and then analyzed by flow cytometry. (**A**) The numbers of hepatic MNCs, and percentages and numbers of hepatic total NK cells (CD3^−^NK1.1^+^). The hepatic total NK cells (CD3^−^NK1.1^+^) were further gated to analyze the percentages of LrNK (CD3^−^NK1.1^+^CD49b^−^CD49a^+^) and cNK (CD3^−^NK1.1^+^CD49b^+^CD49a^−^) cells. The absolute number of LrNK and cNK cells were calculated. (**B**) The numbers of splenic MNCs, and percentages and numbers of splenic total NK cells (CD3^−^NK1.1^+^). The splenic total NK cells (CD3^−^NK1.1^+^) were further gated to analyze the percentages of cNK cells (CD3^−^NK1.1^+^CD49b^+^CD49a^−^). The data are shown as the mean ± SEM. Student’s *t* test was used. No significant statistical difference is defined as ns. * *p* < 0.05.

**Figure 2 viruses-16-00737-f002:**
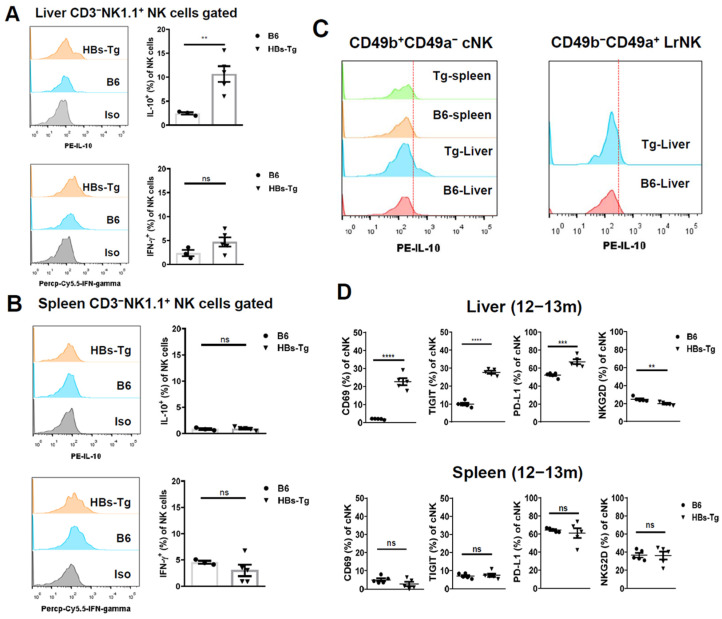
Intrahepatic cNK cells produced high levels of IL-10 and exhibited immune inhibitory features in 12- to 13-month-old HBs-Tg mice. MNCs were isolated from the livers and spleens of 12- to 13-month-old HBs-Tg mice and control WT B6 mice, and then analyzed by flow cytometry. (**A**) IL-10 and IFN-γ expression in total hepatic NK cells (CD3^−^NK1.1^+^). (**B**) IL-10 and IFN-γ expression in total splenic NK cells (CD3^−^NK1.1^+^). The positive percentages of IL-10 and IFN-γ were analyzed according to the histograms, respectively. (**C**) IL-10 expression in hepatic and splenic cNK cells (CD3^−^NK1.1^+^CD49b^+^CD49a^−^) and LrNK cells (CD3^−^NK1.1^+^CD49b^−^CD49a^+^). Representative histograms are shown. The dotted lines were determined according to the isotype control. (**D**) The cNK cells (CD3^−^NK1.1^+^CD49b^+^CD49a^−^) were gated to analyze the expression of phenotypical molecules in the liver and spleen. The data are shown as the mean ± SEM. Student’s *t* test was used. No significant statistical difference is defined as ns. ** *p* < 0.01, *** *p* < 0.001, **** *p* < 0.0001.

**Figure 3 viruses-16-00737-f003:**
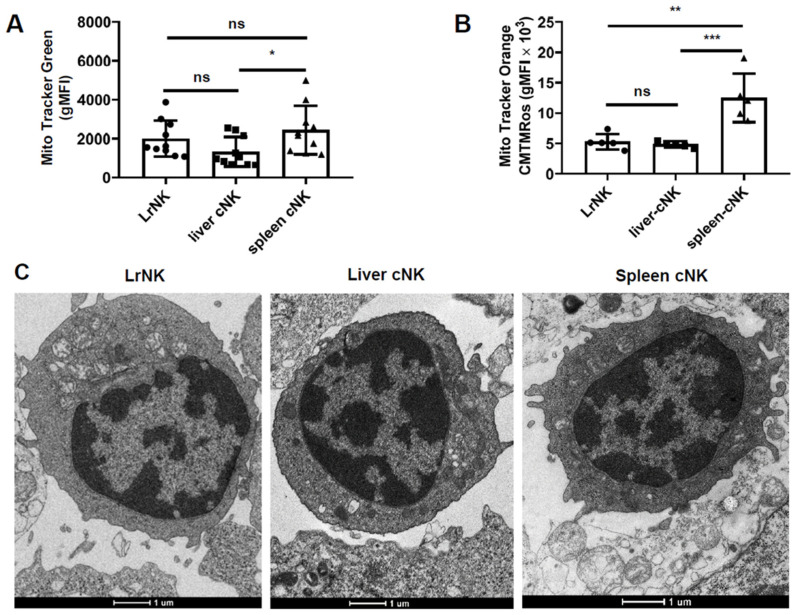
Mitochondrial morphology and function of LrNK, hepatic and splenic cNK cells from HBs-Tg mice. MNCs were isolated from the livers and spleens of 4- to 5-month-old HBs-Tg mice, and then analyzed by flow cytometry. (**A**) Mass of mitochondria determined by MitoTracker^®^ Green FM. (**B**) Membrane potential of mitochondria determined by MitoTracker^®^ Orange CMTMRos. (**C**) Electron microscopic morphology of LrNK, hepatic cNK and splenic cNK cells of 12- to 13-month-old HBs-Tg mice. Ultraslices (50–70 nm) of the cell samples were observed under a transmission electron microscope. The representative ultrastructure is shown. The data are shown as the mean ± SEM. A one-way ANOVA test was used. No significant statistical difference is defined as ns. * *p* < 0.05, ** *p* < 0.01, *** *p* < 0.001.

**Figure 4 viruses-16-00737-f004:**
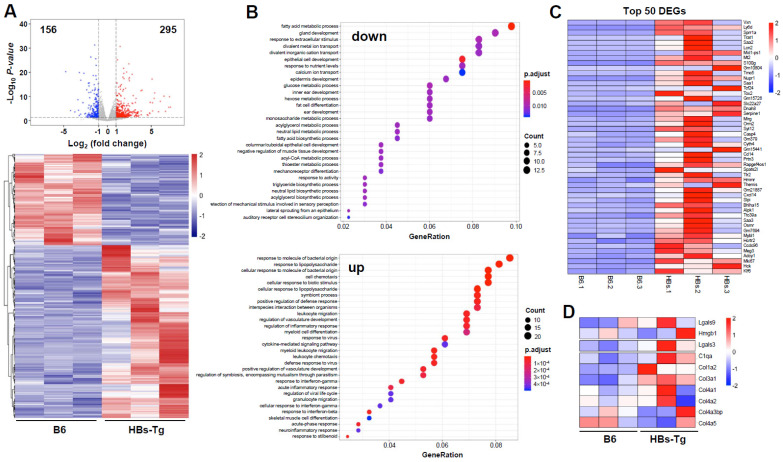
Differentially expressed genes in hepatocytes of HBs-Tg mice compared with WT B6 mice. Hepatocytes were isolated from 4-month-old HBs-Tg mice and control B6 mice, and then analyzed through mRNA sequencing. (**A**) The Volcano plots, based on the fold change and *p* value, show the differential expression of the indicated genes. The two vertical lines correspond to a two-fold change in the expression. The horizontal line indicates *p* = 0.05. The red plots represent the upregulated genes. The blue plots represent the downregulated genes. (**B**) GO enrichment analysis of the DEGs. Biological pathways for downregulated DEGs and upregulated DEGs are shown, respectively. (**C**) Top-50 DEGs are shown. (**D**) Representative DEGs associated with cell adhesion are shown.

**Figure 5 viruses-16-00737-f005:**
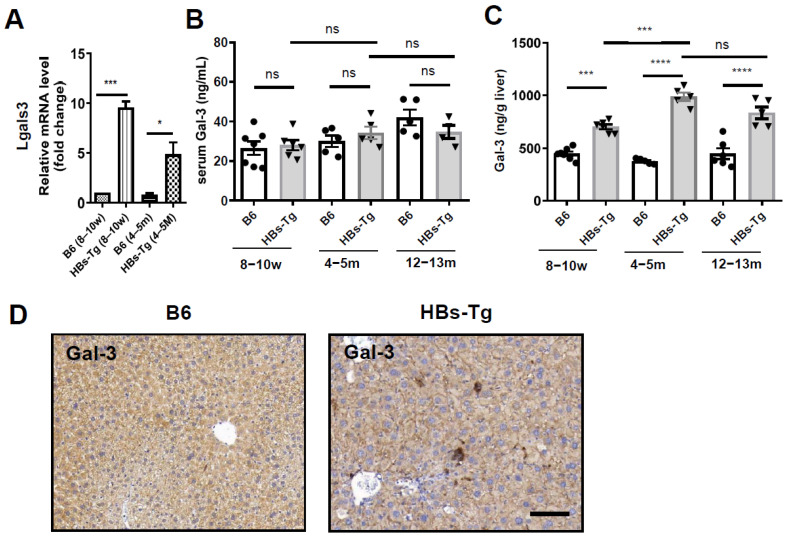
High expression levels of Galectin-3 in HBsAg^+^ hepatocytes. Liver and serum samples were collected from 8- to 10-week-old, 4- to 5-month-old and 12- to 13-month-old HBs-Tg mice and control WT B6 mice, respectively. (**A**) mRNA expression levels of Lgals3 in livers were detected by quantitative real-time PCR. (**B**) Serum levels of galectin-3 (Gal-3) protein detected by ELISA. (**C**) Levels of Gal-3 protein in liver tissues detected by ELISA by using liver homogenate. (**D**) Liver samples were collected from 4- to 5-month-old HBs-Tg mice and control WT B6 mice for Gal-3 staining by IHC. Dark-brown dots represent positive staining. Bar = 100 μm. Data are shown as the mean ± SEM. One-way ANOVA test was used. No significant statistical difference is defined as ns. * *p* < 0.05, *** *p* < 0.001, **** *p* < 0.0001.

**Figure 6 viruses-16-00737-f006:**
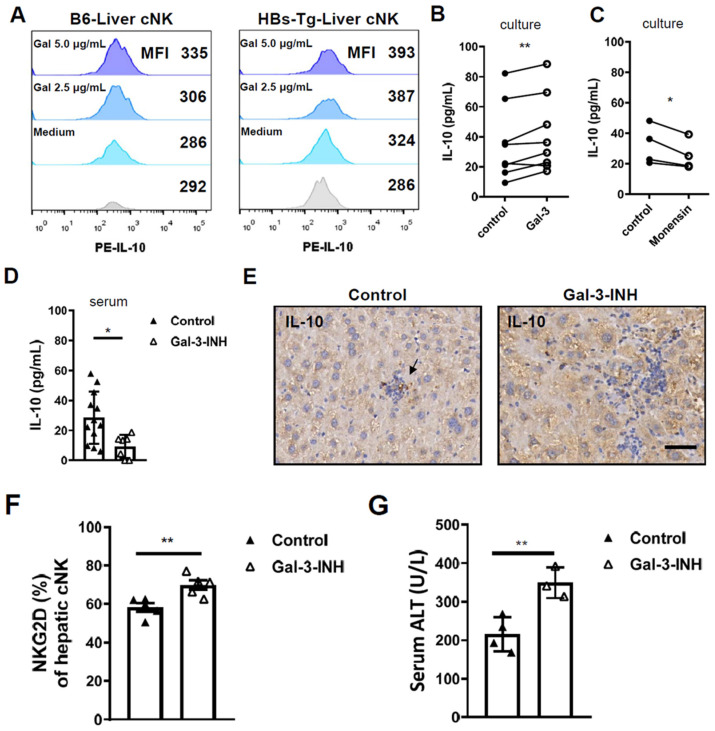
Galectin-3 induced IL-10 production in hepatic cNK cells and prevented hepatocyte damage in HBs-Tg mice. MNCs were isolated from livers of 4- to 5-month-old HBs-Tg mice and control WT B6 mice. (**A**) IL-10 expression in hepatic cNK cells stimulated by rmGal-3. Liver MNCs (5 × 10^5^) were stimulated by recombinant mouse Gal-3 (2.5 μg/mL or 5 μg/mL) in 200 μL of 10% FBS 1640 medium for 48 h. Cultured MNCs were harvested and analyzed by flow cytometry. cNK cells (CD3^−^NK1.1^+^CD49b^+^CD49a^−^) were gated to analyze IL-10 expression. Histograms and MFI are shown. (**B**) Levels of IL-10 protein in culture supernatant. Liver MNCs (5 × 10^5^) were stimulated by recombinant mouse Gal-3 (50 ng/mL) in 400 μL of 10% FBS 1640 medium for 48 h. Culture supernatant was collected and detected by IL-10 CBA flex set. (**C**) Monensin (2.5 μg/mL) was used to block secretion of IL-10 by cultured NK cells in vitro. Paired Student’s *t* test was used. * *p* < 0.05, ** *p* < 0.01. TD139 (Gal-3-INH) (15 μg/g body weight, once every 24 h for three times) was used. (**D**) Serum levels of IL-10 in HBs-Tg mice after TD139 (Gal-3-INH) treatment detected by ELISA. (**E**) IL-10 expression in liver tissues detected by IHC. Dark-brown dots represent positive staining. Bar = 50 μm. (**F**) Hepatic cNK cells (CD3^−^NK1.1^+^CD49b^+^CD49a^−^) were gated and analyzed for expression of NKG2D. (**G**) Serum levels of ALT after TD139 (Gal-3-INH) treatment. Data are shown as mean ± SEM. Unpaired Student’s *t* test was used. * *p* < 0.05, ** *p* < 0.01.

**Figure 7 viruses-16-00737-f007:**
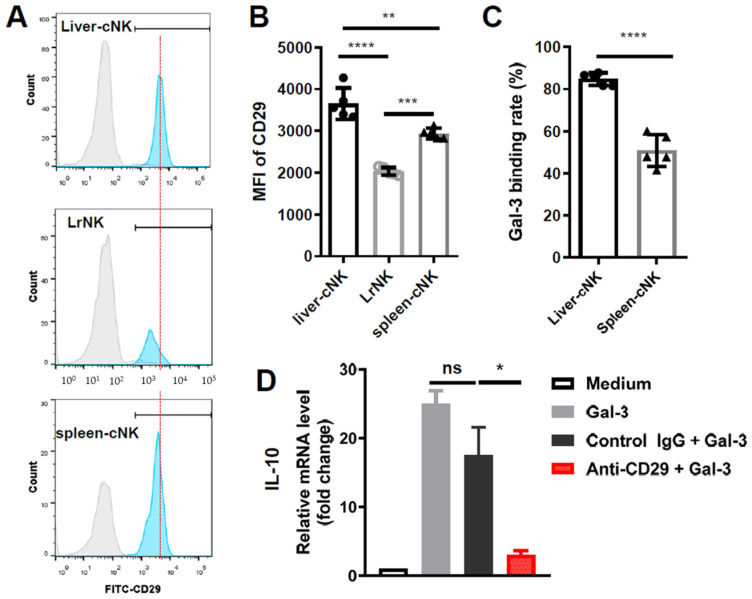
Galectin-3 induced IL-10 transcription via ITGB1 signaling in hepatic cNK cells in HBs-Tg mice. MNCs were isolated from livers and spleens of 12- to 13-month-old HBs-Tg mice. LrNK cells (CD3^−^NK1.1^+^CD49b^−^CD49a^+^) and cNK cells (CD3^−^NK1.1^+^CD49b^+^CD49a^−^) were gated to analyze expression of ITGB1 (CD29) by flow cytometry analysis. (**A**) Representative histograms of CD29 expression. (**B**) MFI of CD29 expression. (**C**) Binding rate of Gal-3 with hepatic cNK cells compared with splenic cNK cells in HBs-Tg mice. (**D**) Relative mRNA expression levels of IL-10 in hepatic cNK cells. Hepatic cNK cells from the livers of 4- to 5-month-old HBs-Tg mice were purified by MACS. cNK cells (1 × 10^5^) were plated and then stimulated with recombinant mouse Gal-3 (2.5 μg/mL) for 48 h. Anti-CD29 mAb (30 μg/mL) was used to block interaction between Gal-3 and CD29. Hamster IgG (30 μg/mL) was used as control. Data are shown as mean ± SEM. One-way ANOVA or Student’s *t* test was used. No significant statistical difference is defined as ns. * *p* < 0.05, ** *p* < 0.01, *** *p* < 0.001, **** *p* < 0.0001.

**Figure 8 viruses-16-00737-f008:**
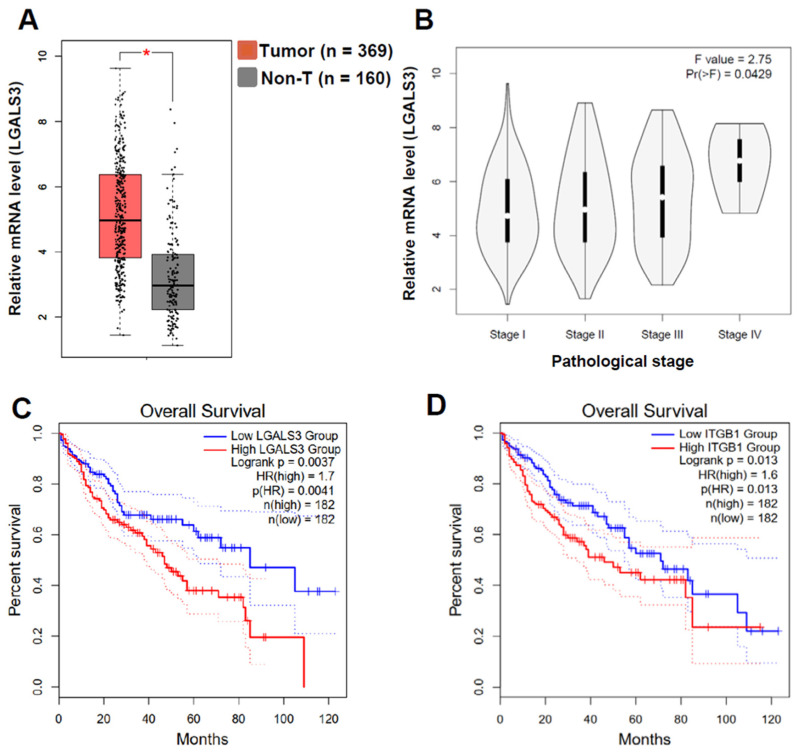
LGALS3 and ITGB1 expression correlated with poor progression and survival of LIHC patients. Gene Expression Profiling Interactive Analysis (GEPIA2) was used to perform analysis for given sets of TCGA expression data. (**A**) Box plots of LGALS3 gene expression in liver hepatocellular carcinoma (LIHC). TCGA normal and GTEx data were matched. Log2 (TPM + 1) was used for log-scale. * *p* < 0.05. (**B**) Pathological stage plots of LGALS3 gene expression in LIHC. Use major stages for plotting. Log2 (TPM + 1) was used for log-scale. (**C**) Overall survival analysis of LIHC patients stratified by LGALS3 gene. (**D**) Overall survival analysis of LIHC patients stratified by ITGB1 gene. Median group cutoff was used. Hazards ratio was calculated based on Cox PH Model; 95% confidence interval (CI) was added as a dotted line.

## Data Availability

The original contributions presented in the study are included in the article/Appendix A, further inquiries can be directed to the corresponding authors.

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
