# Peer review of "Galectin-3-ITGB1 Signaling Mediates Interleukin 10 Production of Hepatic Conventional Natural Killer Cells in Hepatitis B Virus Transgenic Mice and Correlates with Hepatocellular Carcinoma Progression in Patients"

_viruses, 2024, doi:10.3390/v16050737_

Round 1
Reviewer 1 Report
Comments and Suggestions for Authors
Review Report
Galectin-3-ITGB1 signaling mediates IL-10 production of hepatic cNK cells in HBs-Tg mice and correlates with HCC progression in patients by Yongyan Chen et al. Authors have investigated the functional heterogeneity of two different types of intrahepatic NK cells, conventional cNK cells and liver resident NK cells, LrNK cells in a transgenic mouse model of HBsAg. Authors have demonstrated that cNK cells (CD49b+) and not LrNK cells (CD49a+), account for the higher number of total NK cells in the in the liver of HBsAg+ mice during aging and are also responsible for IL-10 production and increased expression of CD69, TIGIT and PD-L1 and lower NKG2D expression in HBs-Tg mice. Authors have also shown that enhanced galectin-3 (Gal-3)-secreted from HBsAg+ hepatocytes accounts for the IL- 10 production of hepatic cNK cells via ITGB1 signaling. In human, LGALS3 and ITGB1 expression positively correlated with IL-10 expression, which in turn correlated with the progression of HCC clinically (OS).
Earlier, LGALS3 has been reported to be a potential therapeutic target in HBV related HCC.
Gal-3-ITGB1 signaling shaped hepatic cNK cells during chronic HBV infection, that may correlate with HCC progression.
Authors should address, the following query.
1. Alessandra Zecca et al 2023, has reported that infiltration of CD49a+ NK cells are associated with poor clinical outcome Iin HCC. Please see the references below.
https://www.ncbi.nlm.nih.gov/pmc/articles/PMC4355863/
https://pubmed.ncbi.nlm.nih.gov/37180150/
https://www.ncbi.nlm.nih.gov/pmc/articles/PMC6274919/

Author Response
Thank you for the comments. As commented, high CD49a+ NK cell infiltrate is associated with poor clinical outcomes in hepatocellular carcinoma (Alessandra Zecca et al. Heliyon. 2023 Dec; 9(12): e22680.). We discussed this issue in the revised manuscript. “Recently, CD49a+ NK cells were reported to be enriched in HCC and a more abundant infiltration was present in patients at advanced stages (35).” was added (Line 400-402).
Reference (https://www.ncbi.nlm.nih.gov/pmc/articles/PMC4355863/) is the paper “Prognostic Significance of Serum Galectin-3 Levels in Patients with Hepatocellular Cancer and Chronic Viral Hepatitis” (Saudi J Gastroenterol. 2015 Jan-Feb; 21(1): 47–50.). We checked the references in the manuscript and found that it was the ref 28 cited in the manuscript (Line 83). Thus, it remains in its original state.
Reference (https://pubmed.ncbi.nlm.nih.gov/37180150/) is the paper “The necroptosis related gene LGALS3 can be used as a biomarker for the adverse progression from chronic HBV infection to HCC” (Front Immunol. 2023 Apr 26:14:1142319). We added this in the discussion of the revised manuscript as followed “After HBV persistent infection, Gal-3 was demonstrated to play a key role in adverse disease progression and be a potential therapeutic target (43).”(Line 427-429).
Reference (https://www.ncbi.nlm.nih.gov/pmc/articles/PMC6274919/) is a review “Natural Killer Cell Dysfunction in Hepatocellular Carcinoma: Pathogenesis and Clinical Implications” (Int. J. Mol. Sci. 2018, 19, 3648). We discussed the functions of NK cells in HBV infection and HBV-related liver diseases in details by citing original articles, so we did not repeat the citation of this review.
Reviewer 2 Report
Comments and Suggestions for Authors
Yongyan Chen et. al. discuss in their article the role of a specific biochemical factor in the progress of a hepatocellular carcinoma, Galectin-3 which binds at the Integrin β-1 receptor.
The outcome of chronic hepatitis B virus infection depends on the quality and quantity of the host immune reaction. A permanent activation of the immune system can contribute to the ongoing destruction of the liver tissue and can lead to the development of a hepatocellular carcinoma. The activation of natural killer cells is part of the immune system response.
Biochemically two types of natural killer cells can be distinguished, conventional natural killer cells and liver-resident natural killer cells. The conventional natural killer cells can secrete large amounts of IFN-γ which is part of their anti-viral activity. The IFN-γ levels correlate with damage to the liver tissue.
Galectin-3 is a biochemical factor which can be produced by tumor cells to protect themselves against an attack by natural killer cells. It is part of the tumor's immune system evasive behaviour.
In this manuscript the authors discover in a mouse model for chronic hepatitis B virus infection with a subsequent hepatocellular carcinoma that Galectin-3 produced by specific hepatocytes induces in the conventional hepatic natural killer cells the production of interleukin-10. Interleukin-10 is immune-suppressive - its production a protective mechanism of the liver tissue.
The authors show in a convincing way that in their mouse model the number of conventional natural killer cells in the liver are increased - and not the resident natural killer cells or the killer cells in the spleen (figure 1).
Only the conventional natural killer cells in the liver are able to produce higher levels of interleukin-10 (figure 2).
Immunohistochemical staining for Galectin-3 reveals that hepatocytes from the mouse model produce significantly more Galectin-3 that in control mouse (figure 5).
Galectin-3 inhibition leads to much lower levels of interleukin-10 production (figure 6) which allows the reverse conclusion that Galectin-3 is responsible for the increase interleukin-10 production in the liver. High interleukin-10 levels is immune-suppressive and protective for the liver tissue.
Galectin-3 can bind to a variety of receptors - but only one of the is over expressed in normal natural killer cells in the liver tissue of the mouse model, the ITGB1 receptor (figure 4). Using an antibody against this receptor leads to a strong suppression of the interleukin-10 production (figure 7).
In their final paragraph the authors relate the expression levels of Galectin-3 and its receptor ITGB1 to the overall survival rate of patients with an hepatocellular tumor and find them negatively correlated. So either the immuno-suppressive effect of interleukin-10 protects the liver tissue but allows progression of the viral infection or the activation of this pathway is a general sign of the severity of the disease which is associated with a lower survival rate.
The authors built within a complex scenario of immune reactions during a chronic hepatitis B virus infection in mouse model a specific cause - effect chain which ultimately leads to interleukin-10 production. The experimental observations are clearly presented and discussed. The data is in agreement with the authors conclusions. I think they successfully described an aspect of this disease. Even if the entire picture might be more complex, working out such cause-effect chains contributes to the understanding of the entire picture.
I think the article can be published as it is.
Author Response
Thank you for the acceptance of the work.